# A Study of Constitutive Model of Rock Damage under the Joint Effect of Load and Moisture

**Zhongguang Sun** [1,*] **, Qinghua Zhang** [2,3,*] **, Jun Zhang** [1] **, Yi Zhang** [1] **and Pengfei Wang** [4]

1 China Coal Technology and Engineering Group Chongqing Research Institute, Chongqing 400039, China
2 School of Energy and Safety, Anhui University of Science and Technology, Huainan 232001, China
3 State Key Laboratory of The Gas Disaster Detecting, Preventing and Emergency Controlling, Chongqing 400037, China
4 College of Mining Technology, Taiyuan University of Technology, Taiyuan 030024, China
* Correspondence: sunzhongguang126@126.com (Z.S.); zqhcqmky@163.com (Q.Z.)

**Abstract:** To study the mechanical damage characteristics of rock under the effect of subversion, a series of mechanical experiments, including both uniaxial and triaxial mechanical compression experiments under various levels of water content were performed. In this study, researchers investigate the impact of water content on the mechanical characteristics of rock, based on the compliance of the rock damage variants to the Weibull statistic distribution, and Drucker–Prager strength rule, aiming to construct a constitutive model under the joint effect of load and moisture. In addition, the established constitutive model is tested in the experiment. According to the test results, during the initial phase of the submersion, the water content in the rock increases following the exponential function until reaching the threshold. The water content remains stable after the threshold. Under the uniaxial and triaxial loads, the damage detected in the rock and the elasticity modulus decreases linearly as the water content increases. The rock's mechanical parameters and the damage evolution rate are significantly impacted by the surrounding pressure. As the surrounding pressure increases, the weakening effect of the water in the rock decreases. The theoretic curves developed to describe the rock damage under the joint effect of the water and load are consistent with the curves drafted based on the test, indicating that the constitutive model can accurately describe the stress and strain behaviors of rock under various water contents and loading conditions.

**Keywords:** sandrock containing water; loading condition; stress–strain behavior; damager constitutive model

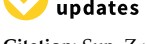



## 1. Introduction

Rock, as one of the common natural materials in engineering projects, often is featured with various degrees of internal macrostructure flaws. Rock failure tends to occur when rock damage continues to accumulate, exceeding the load-bearing capability [1,2]. It is common in the natural environment that rock is surrounded or submerged in water. Water can change the macrostructure of rock through physical, chemical, and mechanical interactions, resulting in decreases in strength, and geological disasters [3–5]. Therefore, studying rock damage variances under the joint effect of load and water content through the constitutive model can provide insightful guidance to engineering project development and implementation [6–8].

The interaction between rock and rock was first proposed by A. M. O Bynhhnkob from the Soviet Union. Since then, the study of water and rock interaction has been transformed into multi-field and interdisciplinary, especially regarding the physical and mechanical characteristics of rock [9–12]. In the mechanical study of rock, compared with the study of elasticity and the study of plastics, the damage theory was launched at a later time. In the year 1958, Kachanov first introduced some basic concepts in the study of rock damage, such as the effective stress and continuous factors, marking the beginning of rock damage

study [13]. In the year 1976, J.W. Dougill first applied damage mechanics to rock study [14], followed by Lemaitre, Chaboche, Krajcinove, Taehyo, Murakami, and Heping Xie, who studied the damage mechanics of rock by taking advantage of different definitions of damage variables, damage parameters, yield principles, and conditions. Now, damage theory has become one of the effective approaches to the study of rock [15–26].

Rock damage mechanic behaviors can be described using the damage variables and the constitutive formulas, which have been studied intensively by domestic and overseas scholars. The constitutive model based on the statistic damage theory has been consistently proven effective. Commonly used probability distribution functions for rock damage models based on statistical theory include the Weibull distribution function, normal distribution function, power function, etc. The statistical damage constitutive model of structure has been proven to be effective in fitting rock constitutive relations and describing the process of rock deformation and failure [27–31]. For example, Erguler Z.A. and Ulusay, R. [32] quantified the effects of water content on the mechanical properties of rocks and developed a method for estimating rock strength and deformability based on physical properties. Cao Wengui and his team [29,33] established a rock statistical damage constitutive model considering the changes of the elastic module, combined with a rock strain softening statistical damage simulation method based on the characteristics of the pore compaction stage. Conil and his team [34] proposed an anisotropic damage model for mudstone based on the Drucker–Prager criterion, which takes into consideration the degradation of the mudstone pore matrix and the changes in hydro-mechanical properties caused by damage. Wang Junbao and his team [35] adopted the Hoek–Brown strength criterion to describe the micro-element strength of rock and deduced the statistical damage constitutive model of rock under three-dimensional stress conditions. Jia Shanpo et al. [36] established a seepage–damage coupling model of mudstone by introducing the damage variables into the seepage–stress coupling control equation. Zhang Xiangdong [37] et al. deduced the modified damage–softening constitutive model of sandstone following a nonlinear fitting method based on the characteristic that rock damage variables comply with the Weibull distribution. Based on the unified strength theory, Hu Xuelong et al. [38,39] established an elastic-plastic damage constitutive model of rock, reflecting the dynamic and static load characteristics of rock. According to the typical triaxial test results of argillaceous sandstone fractured rock mass, Gao Wei et al. [40] proposed a method of establishing the constitutive model suitable for a fractured rock mass in a deep engineering fracture zone. Liu, He, and Cai [41] proposed a damage model featured with the Logistic equation to simulate the stress–strain relation of rocks, which can describe the complete deformation process of rock under uniaxial compression satisfactorily with the simple mathematical function of four model parameters.

In order to describe the stress-strain relationship and the evolution law of rock damage accurately [42], based on the experimental data of loading mechanics, a general expression to describe the effect of rock water content on the weakening of mechanical parameters is proposed in this study. Based on the damage mechanic theory and Weibull statistical distribution theory, the Drucker–Prager criterion is introduced to measure the strength of micro-elements. In addition, a rock damage constitutive model under the joint action of humidity and load is established, which has been verified in the test.

## 2. Damage Constitutive Model of Waterstone

### 2.1. The Mechanical Damage Characteristics of Rock

Considering that the evolution of rock micro-defects is often random, the evolution process of the micro-defect system can be regarded as a non-equilibrium statistical process [2,31], with the following assumptions, including ① the rock materials are macroscopically isotropic; ② the rock micro-units follow Hooke's law; ③ the micro-unit strength follows Weibull distribution, with the probability density function as:

$$\varphi(F) = \frac{m}{F_0}\left(\frac{F}{F_0}\right)^{m-1} exp[-\left(\frac{F}{F_0}\right)^m] \tag{1}$$

In Formula (1) listed above, $F$ is the random distribution variable of Weibull distribution of microelements; m refers to the parameter reflecting rock brittleness; $F_0$ represents the macroscopic average strength of rock; and $\varphi(F)$ refers to a measurement of elemental damage rate during the loading. The damage to the macro elements leads to the macroscopic deterioration of rock specimens.

The damage parameter $D$ is a measure of the damage degree of the material, following the relationship with the probability density of micro-element damage listed below:

$$\varphi(F) = \frac{dD}{dF} \tag{2}$$

$$D = \int_0^F \varphi(F)dF = 1 - exp[-\left(\frac{F}{F_0}\right)^m] \tag{3}$$

According to the equivalent strain assumption, the damage can only affect the strain behavior of the rock through the effective stress, which serves as the foundation of the damage constitutive relationship establishment.

$$\sigma = E_0\varepsilon(1 - D) \tag{4}$$

In Formula (4), $E_0$ is the elastic module of the rock in the dry (no damage) status.

Substituting the damage variable $D$ into Formula (4), the constitutive formula of the sandrock damage can be obtained as follows.

$$\sigma = E_0\varepsilon exp[-\left(\frac{F}{F_0}\right)^m] \tag{5}$$

In Formula (5), the damage principle of the rock follows the Drucker–Prager yield rule, which takes the main middle stress, and the hydrostatic pressure into consideration, resulting in the selection of the micro-strength of the rock.

$$F = f(\sigma) = \alpha I_1 + J_2^{\frac{1}{2}} \tag{6}$$

whereas, $I_1 = \sigma_{ii} = \sigma_1 + \sigma_2 + \sigma_3$, is the first invariant of the stress; $J_2 = \frac{1}{6}[(\sigma_1 - \sigma_2)^2 + (\sigma_2 - \sigma_3)^2 + (\sigma_3 - \sigma_1)^2]$, is the second invariant of the stress; $\alpha = \frac{\tan\phi}{(9+12\tan^2\phi)^{\frac{1}{2}}}$, is constant related to the internal friction angle of rock ($\varphi$ = 37.27°).

(1)    Uniaxial compression status

Under the uniaxial compression, the stress status follows $\sigma = \sigma_1$, $\sigma_2 = \sigma_3 = 0$.

$$F = f(\sigma') = \left(\alpha + \frac{1}{\sqrt{3}}\right)\frac{\sigma}{1 - D(\sigma)} \tag{7}$$

Taking the effective stress into consideration, which allows the substitution of Formula (4) into Formula (7), resulting in:

$$F = f(\sigma) = \left(\alpha + \frac{1}{\sqrt{3}}\right)E_0\varepsilon \tag{8}$$

Substituting Formula (8) into Formula (5), the following formula can be developed.

$$\sigma = E_0\varepsilon e^{-\left(\frac{(\alpha+\frac{1}{\sqrt{3}})E_0\varepsilon}{F_0}\right)^{m1}} \tag{9}$$

$$\frac{d\sigma}{d\varepsilon} = E_0 e^{-\left[\frac{\left(\alpha + \frac{1}{\sqrt{3}}\right)E_0\varepsilon}{F_0}\right]^{m_1}}\left\{1 - m_1\left[\frac{\left(\alpha + \frac{1}{\sqrt{3}}\right)E_0\varepsilon}{F_0}\right]^{m_1}\right\} \tag{10}$$

Considering that in the conventional compression test, in the curve, when $\varepsilon = \varepsilon_{max}$, $\sigma = \sigma_{max}$, $\left.\frac{d\sigma}{d\varepsilon}\right|_{\varepsilon=\varepsilon_{max}} = 0$, the following formula can be obtained.

$$F_0 = \left(\alpha + \frac{1}{\sqrt{3}}\right)E_0\varepsilon_{max}m_1^{\frac{1}{m_1}} \tag{11}$$

$$m_1 = \frac{1}{\ln\left(\frac{E_0\varepsilon_{max}}{\sigma_{max}}\right)} \tag{12}$$

The constitutive formula of rock under the uniaxial compression is obtained:

$$\sigma = E_0\varepsilon e^{-\frac{1}{m_1}\left(\frac{\varepsilon}{\varepsilon_{max}}\right)^{m_1}} \tag{13}$$

Combining Formulas (12) and (13), the mechanical damage of rock due to load can be obtained.

$$D(\sigma) = 1 - e^{-\frac{1}{m_1}\left(\frac{\varepsilon}{\varepsilon_{max}}\right)^{m_1}} \tag{14}$$

(2)　Conventional Triaxial Compression Status

Under triaxial compression status, the stress should follow $\sigma_1 = \sigma \geq \sigma_2 = \sigma_3 \neq 0$. Assuming that $\sigma_3 = f_{31}(\sigma_1)$, following the unloading stress path, the axial confining pressure demonstrates a linear relationship, which is $\sigma_3 = f_{31}(\sigma_1) = K\sigma$, with $K$ as the linear coefficient. Taking the effective stress into consideration, the following formula can be obtained.

$$F = f(\sigma) = \left[\alpha(1 + 2K) + \frac{1}{\sqrt{3}}(1 - K)\right]E_0\varepsilon \tag{15}$$

Substituting Formula (15) into Formula (4), the following formula can be obtained.

$$\sigma = E_0\varepsilon e^{-\left\{\frac{\left[\alpha(1+2K)+\frac{1}{\sqrt{3}}(1-K)\right]E_0\varepsilon}{F_0}\right\}^{m_2}} \tag{16}$$

$$\frac{d\sigma}{d\varepsilon} = E_0 e^{-\left[\frac{\alpha(1+2K)+\frac{1}{\sqrt{3}}(1-K)E_0\varepsilon}{F_0}\right]^{m_2}}\left\{1 - m_2\left[\frac{\alpha(1+2K)+\frac{1}{\sqrt{3}}(1-K)E_0\varepsilon}{F_0}\right]^{m_2}\right\} \tag{17}$$

Considering that during conventional triaxial compression, in the curve, when $\varepsilon = \varepsilon_{max}$, $\sigma = (\sigma_1 - \sigma_3)_{max} = (1 - K)\sigma_{max}$, $\left.\frac{d\sigma}{d\varepsilon}\right|_{\varepsilon=\varepsilon_{max}} = 0$, the following can be obtained.

$$F_0 = \alpha(1 + 2K) + \frac{1}{\sqrt{3}}(1 - K)E_0\varepsilon_{max}m_2^{\frac{1}{m_2}} \tag{18}$$

$$m_2 = \frac{1}{\ln\left[\frac{E_0\varepsilon_{max}}{(1-K)\sigma_{max}}\right]} \tag{19}$$

The constitutive formula under the triaxial compression is obtained:

$$\sigma = E_0\varepsilon e^{-\frac{1}{m_2}\left(\frac{\varepsilon}{\varepsilon_{max}}\right)^{m_2}} \tag{20}$$

Combining Formulas (19) and (20), the mechanical damage of rock due to load can be obtained.

$$D(\sigma) = 1 - e^{-\frac{1}{m_2}\left(\frac{\varepsilon}{\varepsilon_{max}}\right)^{m_2}} \tag{21}$$

*2.2. The Rock Damage Constitutive Model under the Joint Effect of Moisture and Load*

After the rock material absorbs water, the macroscopic mechanical parameters of the rock are weakened. In addition, the rock material after water absorption is affected by the load during the loading process. As a result, the damage of the water-containing sandstone specimen under different stress path conditions should be humidity damage and mechanical damage. The coupling effect can be expressed as:

$$D = D(\omega, \sigma) \tag{22}$$

Since the rock is first weakened by water immersion and then subjected to different stress path correlation tests, it is possible that the damage caused by the two is independent of each other, with the two factors standing alone. According to the equivalent strain assumption, the damage constitutive relation of rock material can be obtained.

$$\sigma = E_0 \varepsilon [1 - D(\sigma)][1 - D(\omega)] \tag{23}$$

## 3. The Mechanical Test of Sandrock Containing Water

### 3.1. Experimental Overviews

Samples were processed into a dimension of $\Phi 50$ mm $\times$ h100 mm, with groups and numbers shown in Figure 1a,b. The mineral composition of this batch of sandstone samples was obtained by D8 ADVANCE X-ray diffractometer analysis, including 35.5% of quartz, 23.4% of plagioclase, 35.9% of calcite, and 5.1% of kaolinite and chlorite, and the test results are shown in Figure 1c.

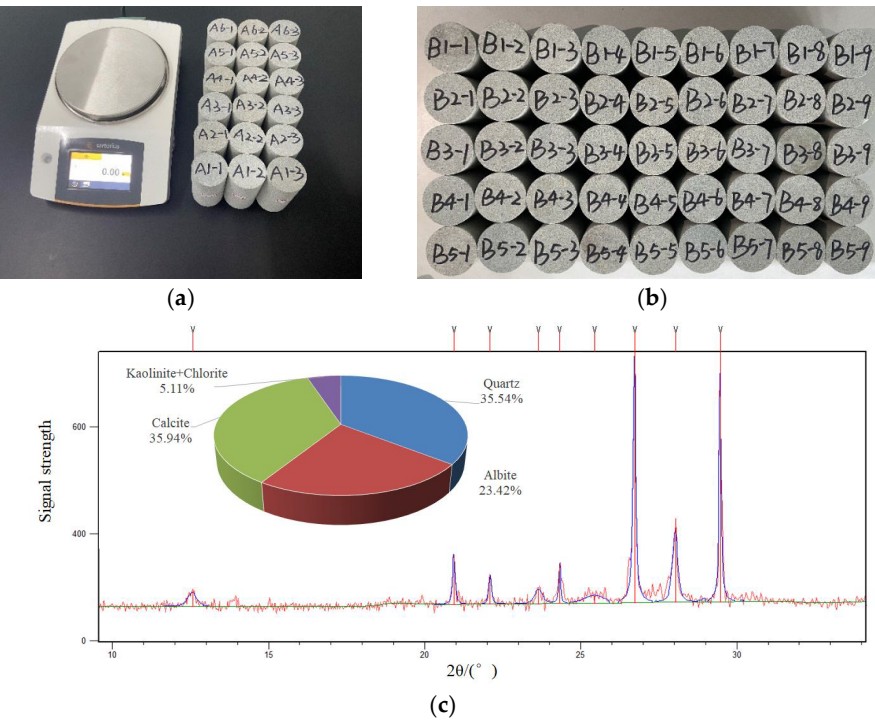

(a)  (b)

(c)

**Figure 1.** Experimental preparation. (**a**) Standard sample used in the uniaxial compression experiment; (**b**) Standard sample used in the triaxial compression experiment; (**c**) Mineral composition of sandstone samples used in the study.

Before the immersion experiment, some tests were performed to obtain basic physical and mechanical properties of the initial state of the rock samples, indicating an average natural water content of 0.45%, and an average natural density of 2.44 g/cm³. The average uniaxial compressive strength is 67.03 MPa. The average compressive strengths under various confining pressure conditions of 10 MPa, 20 MPa, and 30 MPa are 129.08 MPa,

174.99 MPa, and 210.49 MPa, respectively. The calculated internal friction angle of the rock sample is $\varphi = 37.27°$ with a cohesion $C = 22.33$ MPa.

All groups of samples were dried in an XMTD-8222 constant temperature oven, the oven was heated to 105 °C, and maintained at a constant temperature for 48 h, followed by colling till room temperature. The water content of the rock samples was measured after 0.75 h, 1.5 h, 6 h, 24 h, and 720 h of natural soaking.

The mechanical tests of rock samples are all completed by an RMT-301 rock and concrete mechanical test system, as shown in Figure 2. The detailed experimental scheme is as follows:

(1) Single Axis Compression Test: The test samples are numbered from A$i$-$j$, with $i$ = 1~6 (1 refers to the sample in dry status, sample 2 to 6 refer to the samples submerged for 0.75 h, 1.5 h, 6 h, 24 h, and 720 h, respectively) and $j$ = 1~3 (three samples in one test group). The test adopted the displacement control, with the load increased at a rate of 0.5 mm/min until sample failure.

(2) Conventional Three-axis Compression Test: The test samples are numbered from B$i$-$j$, with $i$ = 1~6 (1 refers to the sample in dry status, sample 2 to 6 refer to the samples submerged for 0.75 h, 1.5 h, 6 h, 24 h, and 720 h, respectively) and $j$ = 1~9 (The surrounding pressure is categorized into three sections, with 3 samples in each section). The test adopted the stress control, with the surrounding pressure increased at a rate of 0.05 MPa/s, along with the axial pressure. The initial pressure settings are 10 MPa, 20 MPa, and 30 Mpa, respectively. During the later phases, the control is switched to the displacement control, with a constant surrounding pressure and an increasing axial pressure at a rate of 0.5 mm/min.

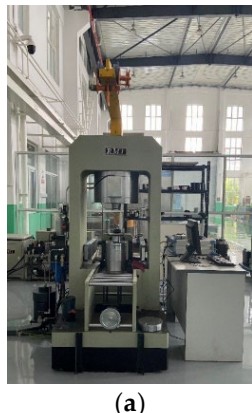 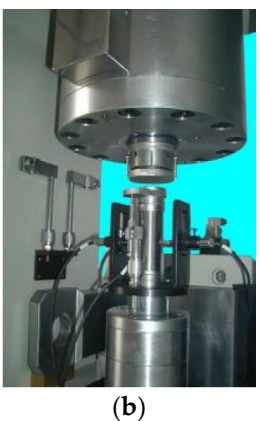 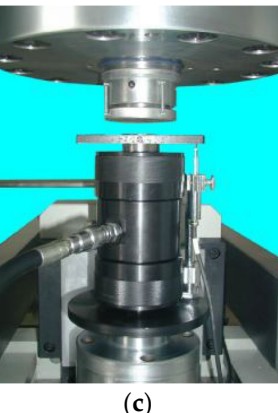

(**a**) (**b**) (**c**)

**Figure 2.** Mechanical experimental equipment. (**a**) Experimental equipment; (**b**) Triaxial compression tester; (**c**) Uniaxial compression tester.

### 3.2. The Water Content Test

After the measurement and calculation, the corresponding relationship between the immersion time $t$ of the rock specimen and the water content $\omega_t$ of the specimen is shown in Figure 3. In the initial stage of water immersion, the water content of the rock specimen increased rapidly, and then the growth rate of water content gradually slowed down. After the water was fully absorbed by the rock, resulting in saturation, the water content of the rock remained at a stable value. Therefore, with a sufficient water supply, the water content of the rock specimen increases with a time threshold $t_0$. In other words, before the time threshold $t_0$ is reached, the corresponding relationship between the immersion time $t$ of the rock specimen and the water content $\omega_t$ follows an exponential growth. After reaching the time $t_0$, the water content of the rock tends to remain constant, reaching the final saturated water content. According to the test results of the water content of rock

samples, the variation law of water content $\omega_t$ with $t$ can be summarized as Formula (24), with the saturated water content of 2.790% after function fitting.

$$\omega_t = \begin{cases} 2.71 - 1.23\mathrm{e}^{-0.26t} & 0 < t < t_0 \\ \omega_{t_0} & t \geq t_0 \end{cases} \tag{24}$$

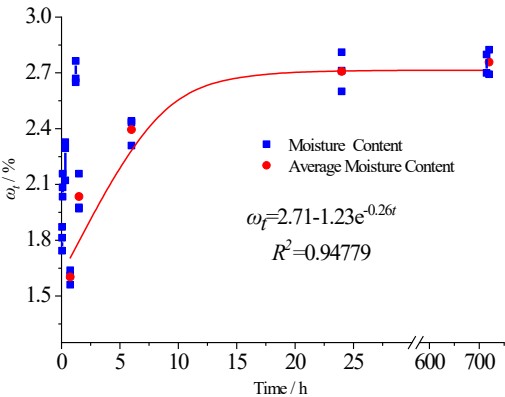

**Figure 3.** The correlation between water content and soaking time.

### 3.3. The Mechanical Test Results of Sandrock Containing Water

As demonstrated in Figures 4 and 5, respectively, the full stress-strain curves of sandstone specimens under different loading stress paths and different water content are plotted based on the experimental data. The statistics data of rock mechanical parameters in uniaxial and triaxial compression tests are shown in Tables 1 and 2.

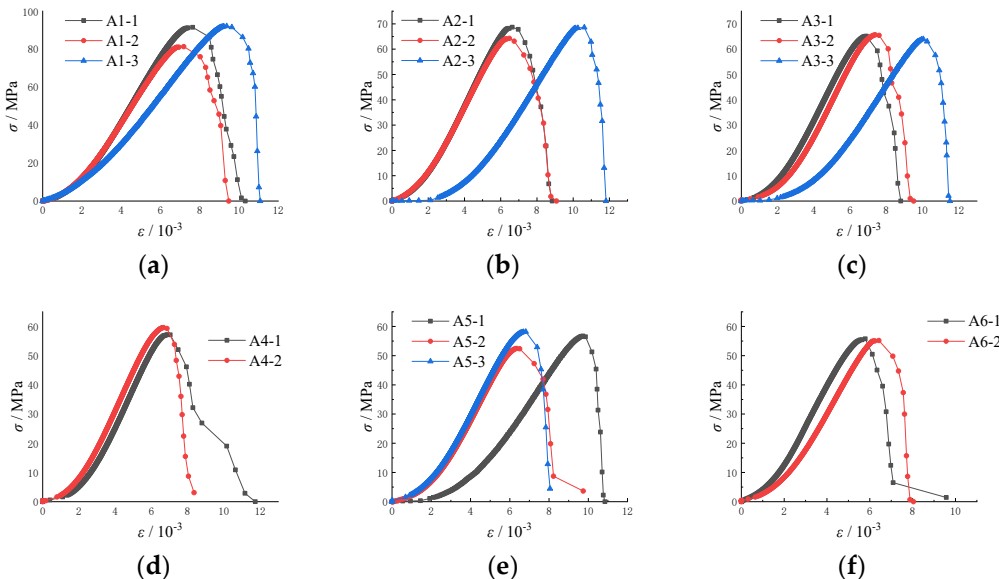

**Figure 4.** The uniaxial compression test results of sandrock containing water. (**a**) Dry Sample; (**b**) Sample Submerged for 0.75 h; (**c**) Sample Submerged for 1.5 h; (**d**) Sample Submerged for 6 h; (**e**) Sample Submerged for 24 h; (**f**) Sample Submerged for 720 h.

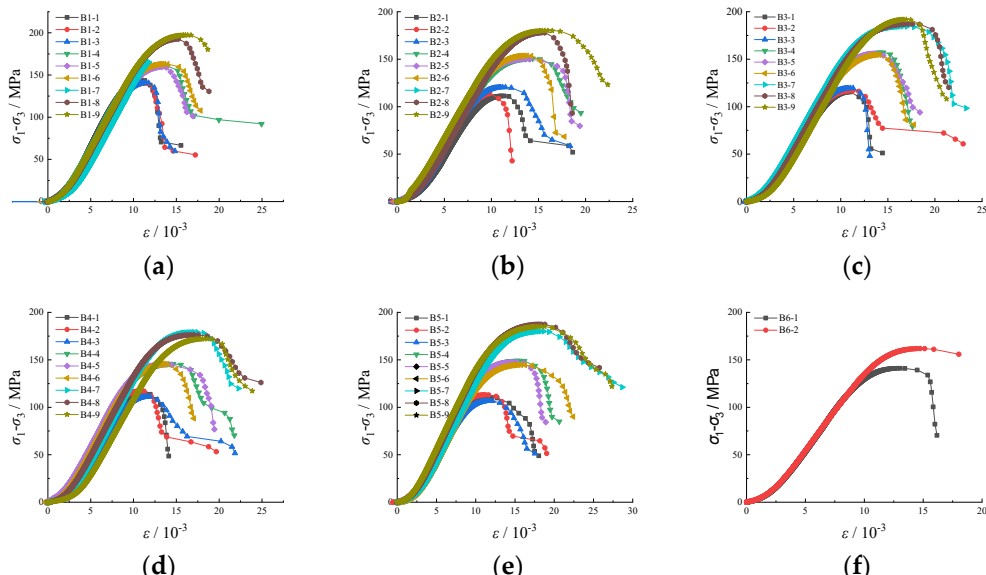

**Figure 5.** The triaxial compression test results of sandrock containing water. (**a**) Dry Sample; (**b**) Sample Submerged for 0.75 h; (**c**) Sample Submerged for 1.5 h; (**d**) Sample Submerged for 6 h; (**e**) Sample Submerged for 24 h; (**f**) Sample Submerged for 720 h.

**Table 1.** Statistics of rock mechanical parameters in uniaxial compression tests.

|  | $t$/h | $\omega_t$/% | $\sigma_c$/MPa |  | $t$/h | $\omega_t$/% | $\sigma_c$/MPa |
|---|---|---|---|---|---|---|---|
| **A1-1** | Dry | 0 | 91.789 | **A4-1** | 6 | 2.742 | 57.172 |
| **A1-2** | Dry | 0 | 81.508 | **A4-2** | 6 | 2.709 | 59.629 |
| **A1-3** | Dry | 0 | 92.376 | **A4-3** | 6 | 2.834 | 57.153 |
| **A2-1** | 0.75 | 1.611 | 68.809 | **A5-1** | 24 | 2.811 | 56.808 |
| **A2-2** | 0.75 | 1.561 | 64.340 | **A5-2** | 24 | 2.960 | 52.465 |
| **A2-3** | 0.75 | 1.638 | 68.704 | **A5-3** | 24 | 2.912 | 58.207 |
| **A3-1** | 1.5 | 1.977 | 65.309 | **A6-1** | 720 | 2.692 | 55.681 |
| **A3-2** | 1.5 | 1.970 | 65.609 | **A6-2** | 720 | 2.824 | 55.135 |
| **A3-3** | 1.5 | 2.158 | 68.540 | / | / | / | / |

According to Figure 4 and Table 1, the comparison of the stress and strain values of each specimen group indicates that the average uniaxial compressive strength of each group of rock specimens decreases significantly with the increase in water content. In addition, the variation of the strain value $\varepsilon$ corresponding to the yield point of each group of rock specimens is not obvious.

As shown in Figure 5 and Table 2, the stress-strain curves of each group of specimens can be divided into three levels based on different confining pressures, indicating that the increase of confining pressure can enhance the specimens. In addition, under the same confining pressure conditions, with the increase of water content, the failure strength of the sandstone specimens significantly decreased, indicating that a smaller confining pressure leads to a faster decrease of the peak strength.

### 3.4. The Impacts of Water Content on the Mechanical Characteristics of Sandrock

In order to further explore the impacts of the water content on the mechanical characteristics of the sandstone specimens, based on the experimental results, the relationship between the failure strength, elastic modulus, and water content of the specimens was fitted and analyzed, as shown in Figures 6 and 7.

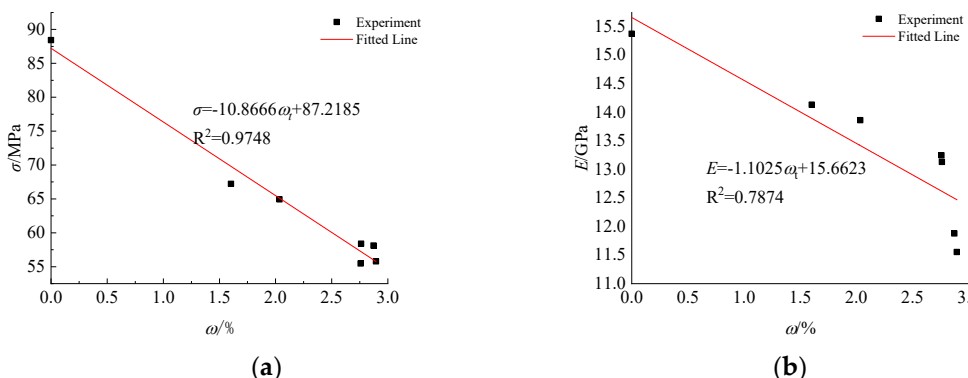

**Figure 6.** The correlation between the water content and the mechanical parameter in uniaxial test. (**a**) $\sigma$-$\omega$; (**b**) $E$-$\omega$.

**Table 2.** The rock mechanical parameters in triaxial compression tests.

| | $t$ h | $\sigma_2 = \sigma_3$ MPa | $\omega_t$ % | $\sigma_c$ Mpa | $E$ Gpa | | $t$ h | $\sigma_2 = \sigma_3$ MPa | $\omega_t$ % | $\sigma_c$ Mpa | $E$ Gpa |
|---|---|---|---|---|---|---|---|---|---|---|---|
| **B1-1** | Dry | 10 | 0 | 154.15 | 20.30 | **B3-7** | 1.5 | 30 | 2.16 | 214.34 | 21.80 |
| **B1-2** | Dry | 10 | 0 | 150.60 | 20.10 | **B3-8** | 1.5 | 30 | 2.03 | 216.48 | 21.79 |
| **B1-3** | Dry | 10 | 0 | 150.88 | 19.87 | **B3-9** | 1.5 | 30 | 2.08 | 222.14 | 21.97 |
| **B1-4** | Dry | 20 | 1.61 | 181.22 | 21.68 | **B4-1** | 6 | 10 | 2.22 | 124.26 | 18.39 |
| **B1-5** | Dry | 20 | 1.56 | 179.99 | 21.51 | **B4-2** | 6 | 10 | 2.21 | 127.32 | 18.55 |
| **B1-6** | Dry | 20 | 1.64 | 183.04 | 22.06 | **B4-3** | 6 | 10 | 2.33 | 121.27 | 17.32 |
| **B1-7** | Dry | 30 | 1.98 | 219.49 | 8.92 | **B4-4** | 6 | 20 | 2.33 | 165.87 | 19.75 |
| **B1-8** | Dry | 30 | 1.97 | 222.95 | 23.38 | **B4-5** | 6 | 20 | 2.30 | 165.25 | 19.89 |
| **B1-9** | Dry | 30 | 2.16 | 227.46 | 23.13 | **B4-6** | 6 | 20 | 2.12 | 165.71 | 20.06 |
| **B2-1** | 0.75 | 10 | 1.17 | 121.27 | 17.28 | **B4-7** | 6 | 30 | 2.04 | 209.42 | 21.42 |
| **B2-2** | 0.75 | 10 | 1.88 | 124.04 | 18.48 | **B4-8** | 6 | 30 | 2.13 | 206.37 | 21.15 |
| **B2-3** | 0.75 | 10 | 1.59 | 130.86 | 18.69 | **B4-9** | 6 | 30 | 2.24 | 202.24 | 20.21 |
| **B2-4** | 0.75 | 20 | 1.87 | 171.26 | 20.51 | **B5-1** | 24 | 10 | 2.62 | 120.25 | 17.39 |
| **B2-5** | 0.75 | 20 | 1.81 | 169.96 | 20.60 | **B5-2** | 24 | 10 | 2.52 | 123.23 | 17.77 |
| **B2-6** | 0.75 | 20 | 1.74 | 174.25 | 21.04 | **B5-3** | 24 | 10 | 2.64 | 116.84 | 17.05 |
| **B2-7** | 0.75 | 30 | 1.75 | 213.35 | 22.10 | **B5-4** | 24 | 20 | 2.60 | 169.22 | 19.89 |
| **B2-8** | 0.75 | 30 | 1.78 | 207.89 | 21.66 | **B5-5** | 24 | 20 | 2.52 | 167.53 | 19.73 |
| **B2-9** | 0.75 | 30 | 1.58 | 210.74 | 21.56 | **B5-6** | 24 | 20 | 2.60 | 164.73 | 19.39 |
| **B3-1** | 1.5 | 10 | 1.97 | 125.76 | 18.24 | **B5-7** | 24 | 30 | 2.65 | 210.15 | 20.66 |
| **B3-2** | 1.5 | 10 | 2.04 | 126.82 | - | **B5-8** | 24 | 30 | 2.67 | 217.90 | 21.70 |
| **B3-3** | 1.5 | 10 | 1.86 | 129.50 | 18.94 | **B5-9** | 24 | 30 | 2.76 | 215.53 | 21.39 |
| **B3-4** | 1.5 | 20 | 1.97 | 177.60 | 20.75 | **B6-1** | 720 | 20 | 2.80 | 158.82 | 20.46 |
| **B3-5** | 1.5 | 20 | 1.93 | 175.70 | 20.88 | **B6-2** | 720 | 30 | 2.70 | 191.88 | 20.84 |
| **B3-6** | 1.5 | 20 | 2.03 | 174.54 | 20.61 | | | | | | |

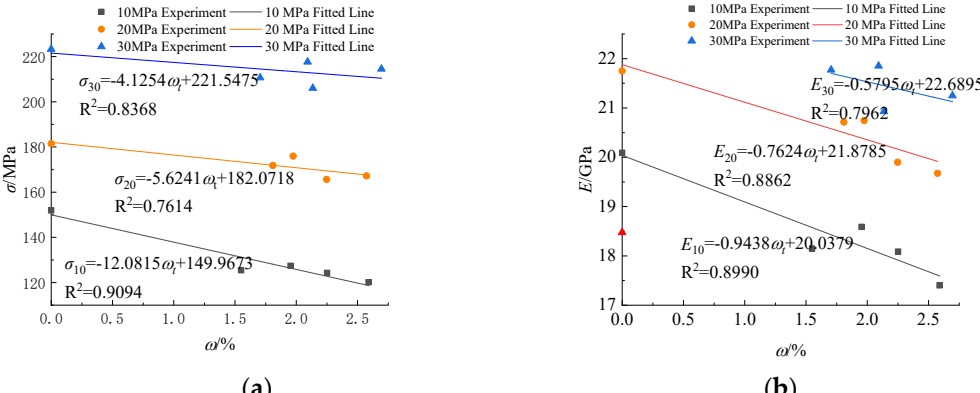

**Figure 7.** The correlation between water content and mechanical parameter in triaxial test. (**a**) $\sigma$-$\omega$; (**b**) $E$-$\omega$.

According to Figure 6, in the uniaxial compression test, with the increase of water content, the failure strength of rock decreases, along with the elastic modulus, demonstrating a relatively clear linear relationship with water content.

According to Figure 7, in the conventional triaxial compression test, the peak strength and elastic modulus of the sandstone under different confining pressures decrease linearly with the increase of water content, demonstrating a higher level of fitting linear relationship under the condition of the low confining pressure. Meanwhile, a lower confining pressure leads to a larger slope of the fitting curve, indicating that the increase in the confining pressure has a positive effect on restraining rock mass damage.

## 4. The Damage Constitutive Model of Water Containing Sandrock Specimens

### 4.1. The Impact of Moisture on the Specimen Damage

According to the previous mechanical tests, under different stress paths, the elastic modulus of sandstone specimens decreases linearly with the increase in water content. Therefore, the elastic modulus $E$ can be used as a damage variable to characterize the effect of water content on the mechanical properties of sandstone specimens. Assuming that the damage value of the dry sandstone specimen is 0, the fitting function of the elastic modulus of the sandstone specimen and the water content is presented below.

$$E = A\omega_t + B \tag{25}$$

In Formula (25), $A$ and $B$ are constants, obtained from experimental statistics.

The elastic modules at different moisture contents are normalized, and the continuity factor can be defined.

$$\omega = \frac{E_{\omega_t}}{E_0} \tag{26}$$

In Formula (26), $E(\omega_t)$ is the elastic module at the water content of $\omega_t$. According to the test results, $E_0 = 15.375$ GPa, $E_{0-10} = 20.090$ GPa, $E_{0-20} = 21.749$ GPa, $E_{0-30} = 18.475$ GPa.

Therefore, the damage in the submerged sandrock specimen is:

$$D(\omega) = 1 - \omega = 1 - \frac{Aw_t + B}{E_0} \tag{27}$$

Based on the test results, the damage variance curve of the water-containing sandstone specimen under different stress paths is drawn as shown in Figure 8. As shown in Figure 8, with the increase in water content, the damage to rock samples increases sharply. When the water content reaches the limit, the damage value reaches 1, with no more accumulation. Meanwhile, with the increase of the confining pressure, the rate of damage evolution slows down, indicating that the ability of rock to resist damage can be strengthened with the increase of confining pressure.

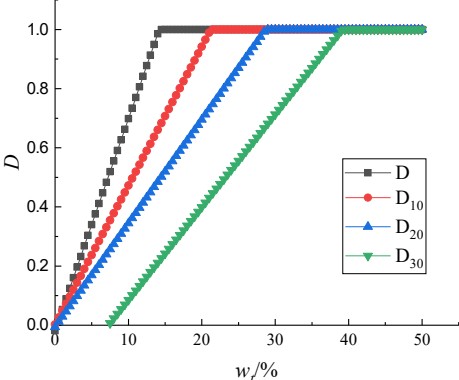

**Figure 8.** The damage variance curve of water-containing sandstone under different stress paths.

### 4.2. The Rock Damage Constitutive Model of Water-Bearing Rock

According to the results of mechanical experiments and Formula (23), the rock damage constitutive model under the joint effect of moisture and load can be deduced, as shown in Formulas (28) and (29).

1.  Uniaxial compression status

$$\overline{\sigma} = \overline{E_0}\,\overline{\varepsilon_{\omega_t}}[1 - D(\sigma)][1 - D(\omega)] = \overline{E_0}\,\overline{\varepsilon_{\omega_t}}\,e^{-\frac{1}{m_1}\left(\frac{\overline{\varepsilon_{\omega_t}}}{\varepsilon_{max}}\right)^{m_1}}\left(\frac{-1.1025\overline{\omega_t} + 15.6623}{\overline{E_0}}\right) \qquad (28)$$

2.  Conventional Triaxial Compression Status

$$\begin{cases} \overline{\sigma_{10}} = \overline{E_{0-10}}\,\overline{\varepsilon_{\omega_t}}\,e^{-\frac{1}{m_2}\left(\frac{\overline{\varepsilon_{\omega_t}}}{\varepsilon_{max}}\right)^{m_2}}\left(\dfrac{-0.5795\overline{\omega_t}+22.6895}{\overline{E_{0-10}}}\right) \\[2ex] \overline{\sigma_{20}} = \overline{E_{0-20}}\,\overline{\varepsilon_{\omega_t}}\,e^{-\frac{1}{m_2}\left(\frac{\overline{\varepsilon_{\omega_t}}}{\varepsilon_{max}}\right)^{m_2}}\left(\dfrac{-0.7624\overline{\omega_t}+21.8785}{\overline{E_{0-20}}}\right) \\[2ex] \overline{\sigma_{30}} = \overline{E_{0-30}}\,\overline{\varepsilon_{\omega_t}}\,e^{-\frac{1}{m_2}\left(\frac{\overline{\varepsilon_{\omega_t}}}{\varepsilon_{max}}\right)^{m_2}}\left(\dfrac{-0.9438\overline{\omega_t}+20.0379}{\overline{E_{0-30}}}\right) \end{cases} \qquad (29)$$

### 4.3. The Verification of Damage Evolution Law of Water-Bearing Rock

Based on the average strength $\overline{\sigma}$ and the average strain ($\overline{\varepsilon_{\omega_t}}$) of the sandstone specimens at different water contents ($\overline{\omega_t}$), the corresponding $m_1$ and $m_2$ values can be obtained, as shown in Tables 3 and 4. The further stress–strain experimental curves and theoretical damage curves of sandstone specimens are shown in Figures 9 and 10, respectively.

(1)  Uniaxial compression status

**Table 3.** The value of $m_1$ under uniaxial compression status.

|  | A1 | A2 | A3 | A4 | A5 | A6 |
|---|---|---|---|---|---|---|
| $t$/h | Dry | 0.75 | 1.5 | 6 | 24 | 720 |
| $\overline{\sigma_c}$/MPa | 89.065 | 67.837 | 65.391 | 58.892 | 56.270 | 54.893 |
| $\overline{E}$/GPa | 15.431 | 13.359 | 12.994 | 13.282 | 11.805 | 13.247 |
| $\overline{\omega_t}$/% | 0 | 1.603 | 2.035 | 2.762 | 2.894 | 2.758 |
| $\overline{\varepsilon_{\omega_t}}$ | 8.627 | 6.795 | 7.643 | 7.313 | 6.100 | 8.627 |
| $\overline{m_1}$ | 2.953 | 1.686 | 1.405 | 1.725 | 1.344 | 1.898 |

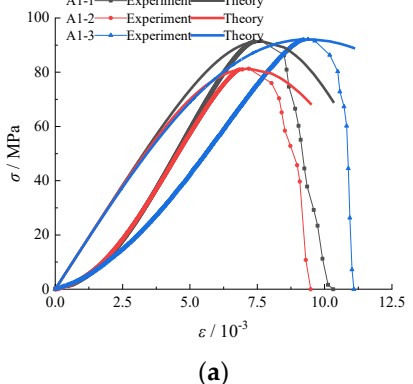

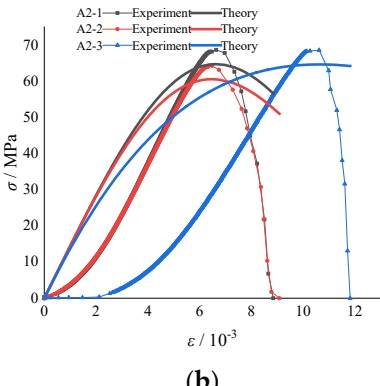

**(a)** **(b)**

**Figure 9.** *Cont.*

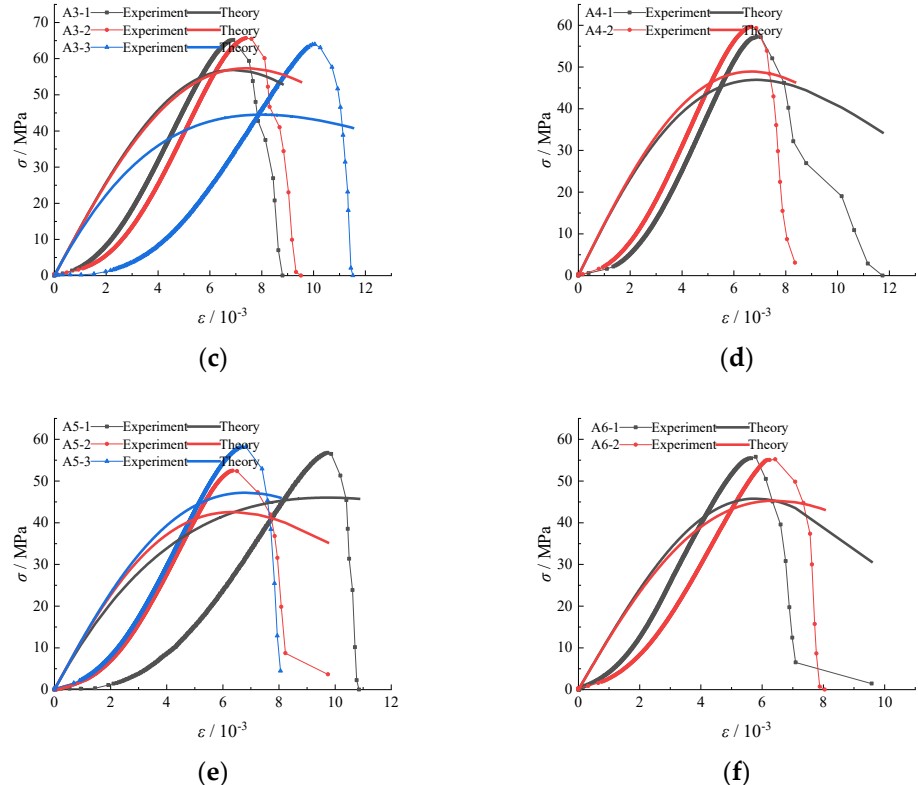

**Figure 9.** The stress-strain experimental curves and theoretical damage curves of sandrock containing water under uniaxial compression status. (**a**) Dry Sample; (**b**) Sample Submerged for 0.75 h; (**c**) Sample Submerged for 1.5 h; (**d**) Sample Submerged for 6 h; (**e**) Sample Submerged for 24 h; (**f**) Sample Submerged for 720 h.

(2) Conventional Triaxial Compression Status

**Table 4.** Value of $m_2$ under triaxial compression status.

|  | **B1** | **B2** | **B3** | **B4** | **B5** | **B6** |
|---|---|---|---|---|---|---|
| $t/h$ | Dry | 0.75 | 1.5 | 6 | 24 | 720 |
| $\overline{\omega_t}/\%$ | 0 | 1.688 | 2.007 | 2.212 | 2.619 | 2.749 |
| $\overline{\varepsilon_{\omega_t-10}}$ | 11.243 | 10.759 | 11.148 | 11.343 | 11.582 | / |
| $\overline{m_{2-10}}$ | 5.184 | 2.911 | 2.713 | 2.393 | 2.097 | / |
| $\overline{\varepsilon_{\omega_t-20}}$ | 13.445 | 13.793 | 14.137 | 13.890 | 15.424 | 13.050 |
| $\overline{m_{2-20}}$ | 3.628 | 3.004 | 2.958 | 2.550 | 2.351 | 2.839 |
| $\overline{\varepsilon_{\omega_t-30}}$ | 15.637 | 15.691 | 16.922 | 17.725 | 18.333 | 14.760 |
| $\overline{m_{2-30}}$ | 4.471 | 3.294 | 2.925 | 2.237 | 2.596 | 2.962 |

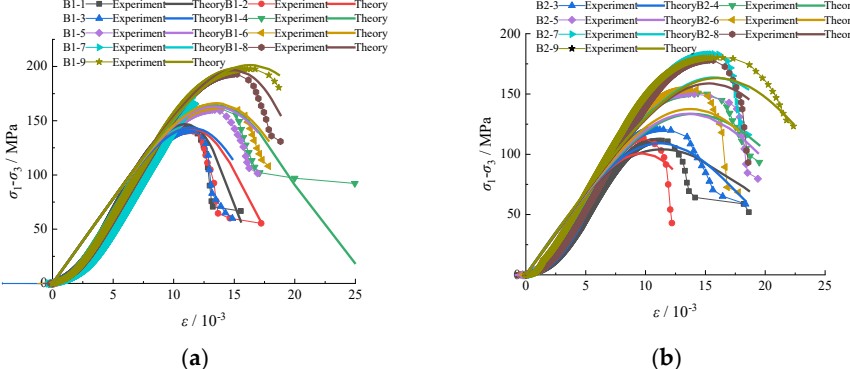

**Figure 10.** *Cont.*

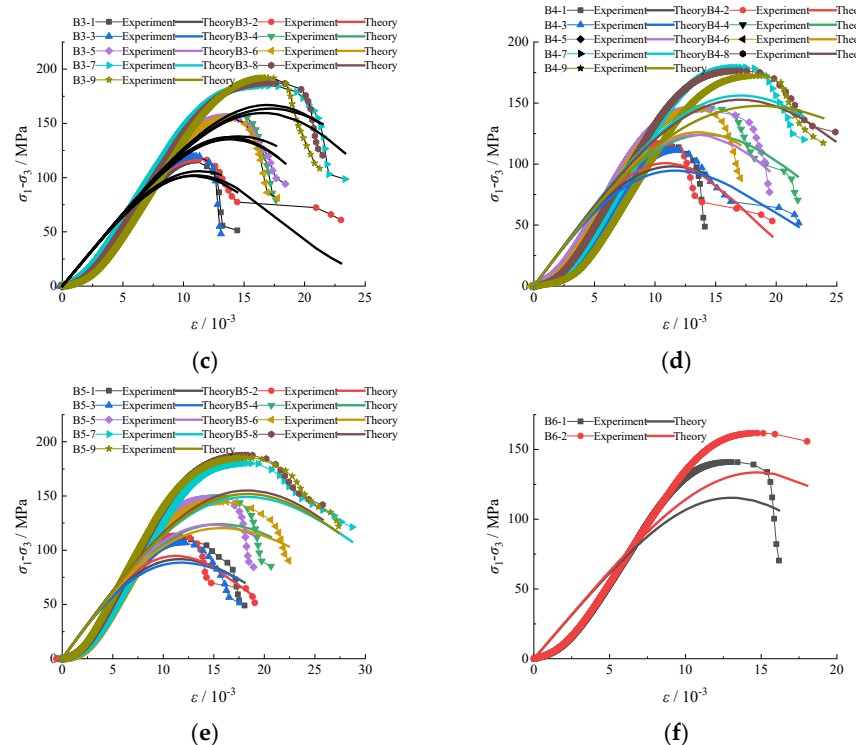

**Figure 10.** The stress-strain experimental curves and theoretical damage curves of sandrock containing water under triaxial compression status. (**a**) Dry Sample; (**b**) Sample Submerged for 0.75 h; (**c**) Sample Submerged for 1.5 h; (**d**) Sample Submerged for 6 h; (**e**) Sample Submerged for 24 h; (**f**) Sample Submerged for 720 h.

According to Figures 9 and 10, under both uniaxial and triaxial loading conditions, the overall shape of the theoretical rock damage curve is similar to the overall shape of the experimental curve, indicating that the theoretical curve of water-containing rock damage evolution can better reflect the sandstone damage under different water contents. Before reaching peak strength, the rock deformation exhibits linear elasticity, and after peak strength, the strength decreases rapidly. With the increase of confining pressure, the theoretical damage curve gradually slows down, which proves that the ability of rock to resist damage increases gradually with the additional increase of confining pressure.

## 5. Discussion

Overall, the theoretical prediction curve derived in this study is consistent with the experimental curve, with some limited deviations, including:

1. The compaction stage of the theoretical damage curve is not obvious, with the curve value higher than the experimental value;
2. The stress of the theoretical damage curve shows a deceleration increase before the peak value, which continues to decrease at an accelerated rate after the peak value, while the stress change rate in the actual compression process is more complicated.

The analysis indicates that the main reasons for the difference are as follows:

① In this study, the researchers assume that the damage value of the dry sandstone specimen is zero (0), and the damage value increases continuously with the accumulation of water content and strain. However, in the compaction stage, the damage value is, in fact, almost unchanged or even reduced, which leads to the stress value of the theoretical curve being larger than the experimental value in the first part of the peak front region.

② Since the theoretical model is a continuously changing function, it is difficult to reflect the localized process in rock failure. Therefore, in order to establish a damage

constitutive model that is more in line with the actual deformation of the rock, the theoretical model needs to be further improved to reflect the actual damage evolution process more accurately.

## 6. Conclusions

Based on the experimental and theoretical analysis results, the following conclusions can be drawn.

1. In the initial stage of immersion, the relationship between the immersion time and the water content of the rock specimen demonstrates exponential growth as a whole. In addition, the water content of the rock remained basically constant after the time threshold $t_0$ is reached.
2. Under both uniaxial compression and conventional triaxial compression, the failure strength and elastic modulus of the sandstone specimens decrease linearly with the increase in water content.
3. As the confining pressure increases, the rate of damage evolution becomes lower, indicating that the increase of the confining pressure tends to reduce the weakening effect of water on rock mass, and demonstrates a positive effect on restraining rock mass damage.
4. The overall shape of the theoretical prediction curve is similar to the loading mechanics curve, indicating that the model can better reflect the stress-strain behavior of sandrock under different water content and uniaxial/triaxial loading conditions.

**Author Contributions:** Conceptualization, project administration, supervision: Q.Z.; investigation and visualization: P.W.; methodology and data curation: J.Z. and Y.Z.; software, validation, formal analysis, writing—review & editing, and writing—original draft: Z.S. All authors have read and agreed to the published version of the manuscript.

**Funding:** This paper was supported by the National Natural Science Foundation of China (52274092), the Regional Innovation and Development Joint Fund of National Natural Science Foundation of China (U21A20110), the Science Innovation and Entrepreneurship Special Funded Projects of Tian Di Science & Technology Co., Ltd. (2022-2-TD-ZD010), the Talent Supported by China Association for Science and Technology (YESS20200211), the Young Talent Programme of Think Tank of China Association for Science and Technology (20220615ZZ07110134), the Natural Science Foundation of Chongqing (cstc2020jcyj-msxmX0793 and cstc2020jcyj-msxmX0972), the Chongqing Talent Program (CQYC20210301417), and the Independence Project of CCTEG Chongqing Research Institute (2020ZDXM06). We thank State Key Laboratory of Coking Coal Exploitation and Comprehensive Utilization for providing the experimental conditions. The authors thank the anonymous reviewers for offering extraordinary advice.

**Institutional Review Board Statement:** Not applicable.

**Informed Consent Statement:** Not applicable.

**Data Availability Statement:** The data used to support the findings of the study are available from the corresponding author upon request.

**Acknowledgments:** The authors appreciate the comments and suggestions by the editors and anonymous reviewers.

**Conflicts of Interest:** The authors declare no conflict of interest.

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
