# Peer review of "A Study of Constitutive Model of Rock Damage under the Joint Effect of Load and Moisture"

_applsci, doi:10.3390/app13021224_

Round 1

Reviewer 1 Report

Dear Authors, I have read the text carefully and have suggestions for a few changes: 

- In the introduction there is a name written in Cyrillic, please change it to English transcription.

- In an international journal, there should be no such sentences: "Overall, limited studies have been conducted on the coupled damage effect of rock mass under water-rock interaction at home and abroad."

- The entire Chapter 4.2. and 4.3. basically describe the known stress propagation in rock and should be moved before Chapter 2. Methodology. 

- I don't understand the title of Chapter 5 very well.

In general, I understand that the infiltration of specimens can affect their engineering properties, but it would also be useful to show what is the effective porosity of these rocks, whether this porosity is oriented and what is its arrangement in relation to the action of forces. These factors can also affect the strength values of the rocks, especially when the pores during tectonic action in the rock mass may be exposed to shear actions.  

Author Response

Dear reviewer,

Thank you for your review and comments on my paper and it's my pleasure.

I have revised it in accordance with the suggestions mentioned, see the red part of the manuscript or the following explanation for details.

(1) In terms of language, it has been polished by a professional service company.

(2) In the introduction there is a name written in Cyrillic, I have changed it to English transcription(|A.M. О Bynhhnkob).

(3) The introduction had been improved, and inappropriate expressions in the introduction were deleted as recommended by experts.

(4) The article structure has been adjusted, and the entire Chapter 4.2. and 4.3. had been moved before Chapter 2.

(5) The title of Chapter 5 has been changed to Discussion.

(6) I understand that the infiltration and porosity of rock specimens can affect their engineering properties, in this paper, we focus on researching the relation between rock strength parameter and the rock damage law by rock mechanic experiments, the effect of microstructure on mechanical properties has not been discussed, we will carry out further research in this field.

Finally, I would like to express my gratitude once again.

All authors as follows: 

Zhongguang SUN *, Qinghua Zhang *, Jun Zhang, Yi Zhang, Pengfei Wang

Reviewer 2 Report

I read with interest the manuscript “A Study of Constitutive Model of Rock Damager under the Joint Effect of Load and Moisture” by Zhongguang Sun et al., submitted to Applied Sciences.

The authors performed several uniaxial and triaxial compressive tests on water-saturated sandstones to investigate the influence of water content on mechanical parameters (mainly compressive strength and elastic modulus) and develop a constitutive model of rock damage.

Overall, I find the manuscript interesting. Particularly, numerous experiments were performed, potentially capable of providing a large and coherent experimental dataset on mechanical properties of a (partially and fully) water-saturated sandstone. However, I feel that numerous and crucial changes should be made in the manuscript:

- English language and style require an extensive editing/improvement. I think that currently they could be inadequate for Applied Sciences;

- The description of methods and presentation of the results (in the Section 2 and 3) are very incomplete. Particularly all the experimental data (failure strength, elastic modulus and water content) should be fully provided in appropriate tables or as supplementary material. In fact, it can be a useful dataset for the scientific community; mostly, often it is not clear what data has been measured and plotted for each sample (especially in figure 3, 6 and 7);

- Actually, I'm no expert on the approaches used to develop the proposed constitutive model. However, it needs to be better introduced and described in the Sections 4. Particularly, literature studies used by the authors to set up the development of their model should be properly cited and introduced/recalled;

- Many references seem poorly accessible to an international audience (often published in Chinese national journals). When possible, they should be integrated/replaced with additional references;

I uploaded a .pdf containing several comments on the issues listed above (as no line numbers were provided in the text).

Author Response

Dear reviewer,

Thank you for your review and comments on my paper and it's my pleasure.

I have revised it in accordance with the suggestions mentioned, see the red part of the manuscript or the following explanation for details.

(1) In terms of language, it has been polished by a professional service company.

(2) The introduction had been improved, and some reference expressions of Chinese scholars in the introduction were substituted, such as references [12], [32],[41],[42].

(3) The article structure has been adjusted, and the entire Chapter 4.2. and 4.3. had been moved before Chapter 2. Literature studies used to set up the development of their model had been cited and introduced/recalled.

(4) All the experimental data (failure strength, elastic modulus and water content) had been provided in appropriate table1 and table 2.

Finally, I would like to express my gratitude once again.

All authors as follows: 

Zhongguang SUN *, Qinghua Zhang *, Jun Zhang, Yi Zhang, Pengfei Wang

Reviewer 3 Report

The paper has been written very well by the authors. However, it is suggested to have an English proof-reading on it. The introduction should also be improved. Please cite the following references in your paper:

1. https://doi.org/10.1088/1742-2140/aacee3

2. https://doi.org/10.1016/j.earscirev.2021.103755

3. https://doi.org/10.1016/j.jappgeo.2014.07.016

4. https://doi.org/10.1016/j.jngse.2016.06.029

Author Response

Dear reviewer,

Thank you for your review and comments on my paper and it's my pleasure.

I have revised it in accordance with the suggestions mentioned, see the red part of the manuscript or the following explanation for details.

(1) In terms of language, it has been polished by a professional service company.

(2) The introduction had been improved, and the following references had been cited in my pape.

[12] Hosseini, M. Estimation of mean pore-size using formation evaluation and Stoneley slowness, J. Nat. Gas Sci. Eng, 2016,33, 898-907.

[42] Hosseini, M. Formation evaluation of a clastic gas reservoir: presentation of a solution to a fundamentally difficult problem. J. Geo. Eng, 2018, 15, 2418–2432.

Finally, I would like to express my gratitude once again.

All authors as follows: 

Zhongguang SUN *, Qinghua Zhang *, Jun Zhang, Yi Zhang, Pengfei Wang